# Solar-Driven Syngas Production Using Al-Doped ZnTe Nanorod Photocathodes

**DOI:** 10.3390/ma15093102

**Published:** 2022-04-25

**Authors:** Youn Jeong Jang, Chohee Lee, Yong Hyun Moon, Seokwoo Choe

**Affiliations:** Department of Chemical Engineering, Hanyang University, Seongdong-gu, Seoul 04763, Korea; chgml9612@hanyang.ac.kr (C.L.); myh4791@hanyang.ac.kr (Y.H.M.); choe5192@hanyang.ac.kr (S.C.)

**Keywords:** photoelectrochemical, CO_2_ reduction, syngas, ZnTe, nanorod, photocathode

## Abstract

Syngas, traditionally produced from fossil fuels and natural gases at high temperatures and pressures, is an essential precursor for chemicals utilized in industry. Solar-driven syngas production can provide an ideal pathway for reducing energy consumption through simultaneous photoelectrochemical CO_2_ and water reduction at ambient temperatures and pressures. This study performs photoelectrochemical syngas production using highly developed Al-doped ZnTe nanorod photocathodes (Al:ZnTe), prepared via an all-solution process. The facile photo-generated electrons are transferred by substitutional Al doping on Zn sites in one-dimensional arrays to increase the photocurrent density to −1.1 mA/cm^2^ at −0.11 V_RHE_, which is 3.5 times higher than that for the pristine ZnTe. The Al:ZnTe produces a minor CO (FE ≈ 12%) product by CO_2_ reduction and a major product of H_2_ (FE ≈ 60%) by water reduction at −0.11 V_RHE_. Furthermore, the product distribution is perfectly switched by simple modification of Au deposition on photocathodes. The Au coupled Al:ZnTe exhibits dominant CO production (FE ≈ 60%), suppressing H_2_ evolution (FE ≈ 15%). The strategies developed in this study, nanostructuring, doping, and surface modification of photoelectrodes, can be applied to drive significant developments in solar-driven fuel production.

## 1. Introduction

Synthetic gas (syngas), a mixture of carbon monoxide (CO) and hydrogen gas (H_2_), is an essential precursor for the production of hydrocarbon fuels and value-added chemicals, such as alcohols and acetic acid [1,2]. Industrial syngas is typically produced from fossil fuels and natural gases at high temperatures and pressures [3,4]. However, employing photoelectrochemical (PEC) water and CO_2_ reduction at ambient temperatures and pressures is a sustainable and environmentally friendly alternative to traditional syngas production [5,6]. 

PEC cells comprising semiconductor photoelectrodes can harvest solar energy to perform electrochemical catalytic reactions [7,8,9]. The standard reduction potential of CO_2_ to CO (CO_2_ + 2H^+^ + 3e^−^→CO + H_2_O, E_o_ = −0.11 V_RHE_) is slightly more negative than that of water to H_2_ (2H^+^ + 2e^−^→H_2_, E_o_ = 0.0 V_RHE_). Therefore, ideally, any photocathode can be used in photoelectrochemical CO_2_ reduction to produce H_2_ in a thermodynamic manner. Moreover, CO_2_ to CO conversion is kinetically poorer than water-to-H_2_ conversion, primarily because of the slow CO_2_ adsorption and intermediate formation [10,11,12]. This indicates that direct syngas production can be achieved using a single photoelectrochemical CO_2_ reduction system. Furthermore, the ratio of H_2_/CO, which is the key factor determining the upgraded product selectivity, is controllable by catalysts [13,14,15]. 

ZnTe semiconductors are an interesting and useful class of photoelectrocatalysts and have received considerable attention owing to their narrow bandgap energy (2.26 eV) and band alignment. This allows harvesting of visible light and a suitable band alignment for syngas production. Specifically, the conduction band minimum (CBM) of ZnTe located at −1.63 V_RHE_, which is highly negative compared to CO_2_-to-CO reduction potential, can provide a strong driving force for photoexcited electron injection for syngas production [16,17,18]. However, ZnTe photocathodes typically exhibit poor photoexcited charge separation because of the rapid charge recombination and kinetically inactive catalytic reaction [16,19].

Therefore, this study proposes Al-doped ZnTe nanorod photocathodes for photoelectrochemical syngas production. The one-dimensional array ZnTes can provide a shortened carrier transport distance, which can facilitate their separation. Furthermore, ZnTes with substitutional Al dopants at Zn sites exhibit increased charges transport activity in the bulk compared with pristine ZnTe. The difference in valence states between Al (Al^3+^ in lattice) and Zn (Zn^2+^ in lattice) causes a significantly increasing major carrier concentration. In addition, PEC syngas production is demonstrated, enabling a change of the H_2_:CO ratio from 5:1 to 1:4, using surface-modified ZnTe-based photocathodes. The proposed method results in a significant enhancement in syngas production and H_2_/CO ratio changes. The photocathode preparation, results, and effects of catalyst modification are presented in this paper.

## 2. Materials and Methods

### 2.1. Materials 

Fluorine doped tin oxide substrate was purchased from Pilkington, Lancashire, UK. Zn(NO_3_)_2_∙6H_2_O (99%), ammonia water (28–30%), Al(NO_3_)_3_∙9H_2_O (98%), Na_2_TeO_3_ (99%), NaBH_4_ (99%) were purchased from Aldrich, Burlington, MA, USA. KHCO_3_ (99.7%) was purchased by Junsei chemicals, Tokyo, Japan. All chemicals used in this work were as received, without further purification.

### 2.2. Synthesis of Al Doped ZnTe Nanorod Film 

Al-doped ZnTe nanorods were prepared using the following two steps. First, the Al doped ZnO nanorods were solvothermally grown on a ZnO (50 nm)-sputtered fluorine-doped tin oxide substrate (FTO, PECTM 8/Pilkington, Lancashire, UK) in an aqueous solution containing 10 mM Zn(NO_3_)_2_∙6H_2_O (Aldrich, Burlington, MA, USA), ammonia water (28–30%, Aldrich, Burlington, MA, USA), and Al(NO_3_)_3_∙9H_2_O (Aldrich, Burlington, MA, USA) at 95 °C for 2 h [19,20]. The concentration of Al precursors was adjusted to 0, 1, 3, and 5 atomic % against the concentration of Zn precursors, and the samples were named as Al:ZnO x, where x = 0, 1, 3, and 5, respectively. Then, Al doped ZnTe nanorods were prepared from Al-doped ZnO nanorods using the previously reported anion exchange reaction in an aqueous solution containing 0.45 mM Na_2_TeO_3_ (Aldrich, Burlington, MA, USA) and 26.5 mM NaBH_4_ at 95 °C for 2 h [16]. Because the Al dopant concentration in ZnTe corresponds to the concentration in ZnO, Al-doped ZnTe nanorod samples were named Al:ZnTe X, where X = 0, 1, 3, and 5, respectively. 

### 2.3. Physico and Chemical Characterizations

The crystal structures of Al-doped ZnO and Al-doped ZnTe were examined using X-ray diffraction (XRD) (Mac Science, Kanagawa, Japan, M18XHF using Cu Ka radiation, λ = 0.15406 nm). The morphologies of the electrodes were investigated using field-emission scanning electron microscopy (FESEM) (JEOL, Tokyo, Japan, JMS-7401F and Phillips Electron Optics B.V. XL30S FEG, operated at 10 keV) and high-resolution transmission electron microscopy (HR-TEM) (JEOL, Tokyo, Japan, JEM-2200FS), combined with an energy dispersive X-ray spectrometer operated at 200 kV. The elemental compositions and their oxidation states were investigated using X-ray photoelectron spectroscopy (XPS) (Thermo Fisher Scientific, Waltham, MA, USA, ESCALAB 250Xi) and the binding energy of each element was calibrated with respect to the carbon 1 s peak at 284.8 eV. The absorbance of photoelectrodes was examined using UV-Vis diffuse reflectance spectroscopy (UV-Vis DRS) (Shimadzu, Kyoto, Japan, UV2501PC).

### 2.4. Photoelectrochemical Measurements

All photoelectrochemical CO_2_ reduction experiments (J-V, J-t, and EIS) were conducted in a three electrodes configuration with ZnTe-based working electrodes, a graphite rod counter electrode, and an Ag/AgCl (4 M KCl) reference electrode, under simulated solar illumination in an undivided gas-tight cell. Simulated solar illumination was generated using a 300 W Xe lamp (Newport, CA, USA, Oriel, 91–160,) with an AM 1.5 G and an IR filter. The light intensity of 100 mW/cm^2^ was calibrated using the guaranteed reference by National Renewable Energy Laboratories, US. A potentiostat (Gramry, Warminster, PA, USA, Reference 600TM) supplied bias to adjust the potential difference to electrodes connected in a circuit. The exposed comparable illuminating area (0.25–0.3 cm^2^ for J-V measurements and 1–1.2 cm^2^ for J-t measurements) of the photocathodes was masked with insulating epoxy. CO_2_-saturated 0.5 M KHCO_3_ (Junsei chemicals, Tokyo, Japan) in water was used as the electrolyte.

J-V measurements were performed while cathodically sweeping the potentials at a scan rate of 10 mV/s. J-t measurements were recorded for 3 h at −0.11 V and −0.3 V with respect to RHE. While performing J-t measurements, the produced gases were analyzed using gas chromatography (Agilent, Santa Clara, CA, USA, Model 7890) with a Carboxen 1000 packed column and a thermal conductivity detector for gaseous products—H_2_, CO, CO_2_, and CH_4_. Electrochemical impedance spectroscopy (EIS) measurements were taken at a constant DC potential of −0.11 V vs RHE and an AC potential frequency range of 0.1–100,000 Hz with a 20 mV amplitude.

The faradaic efficiency (FE) is determined by dividing the number of charges used to produce the detected number of products, H_2_ or CO, by the total charge passed during the photoelectrochemical measurement using the following equation, where n is the number of electrons required to produce one H_2_ or CO molecule, which is 2, and F is Faraday’s constant (96485.33 C/mol).
FE (%) = (n × mol of product × F)/(Total charge passed) × 100

## 3. Results and Discussion

### 3.1. Synthesis of Al:ZnTe Nanorod Photocathodes

The Al-doped ZnTe (Al:ZnTe) nanorod photocathodes used in this work were prepared using a two-step, modified hydrothermal method and following the ion exchange reaction. The preparation method is depicted in the schematic diagram (Appendix A). The Al-doped ZnO (Al:ZnO) nanorods were hydrothermally grown on the FTO with Al-dopant concentrations in the range of 0–5 at% compared with the concentration of Zn precursors. The Al:ZnO nanorods served as an in situ template and precursors to prepare Al:ZnTe. As the solubility product constant (K_sp_) for ZnTe (5.0 × 10^−34^) is significantly lower than that for ZnO (6.8 × 10^−34^), ZnO nanorods were transformed to ZnTe by spontaneous anion exchange reactions and thus ZnO core-ZnTe shell heterojunction type electrode was prepared [21]. 

ZnTe nanorods with different concentrations of Al dopants (Al:ZnTe X, where X = 0, 1, 3, and 5 represent the at% of Al compared with Zn) were prepared and their photoelectrochemical performance was investigated in Appendix A. Amo1ang the photocathodes, Al:ZnTe 3, containing 3 at% Al dopant, exhibited the highest photocurrent densities. The optimized Al:ZnTe 3 photocathodes were named Al:ZnTe and their physicochemical characteristics and photoelectrochemical activities were compared with those of pristine ZnTe photocathodes. 

The XRD patterns of ZnTe and Al:ZnTe showed hexagonal wurtzite ZnO (JCPDS no. 01-089-0511) and zinc blend ZnTe (JCPDS no. 01-065-0385) (Figure 1a). All prepared photocathodes showed two patterns of ZnTe and ZnO, regardless of the Al dopant concentration (Appendix A). Furthermore, neither peak splitting nor shifting was observed by Al doping. Moreover, no crystal structure change occurred by Al doping. 

The presence of Al^3+^ in Al:ZnTe was confirmed by the XPS results for the Al 2p orbital (Figure 1b). Additionally, the oxidation states of Zn and Te elements were analyzed to investigate physicochemical changes for ZnTe by Al doping. In the core level of the Zn 2p spectra, two peaks attributed to Zn^2+^ from ZnO and ZnTe slightly positive shifted 0.5 eV by doping with the electron-deficient Al^3+^ relative to Zn^2+^. However, no significant peak shifts were observed in the core level of the Te 3d results (Figure 1c,d) [22].

Another distinct difference in the morphologies for ZnO and ZnTe resulting from Al doping was observed. The pristine ZnO had a cylindrical nanorod, whereas Al:ZnO had a needle-shaped nanorod (Figure 2a,c). As the concentration of Al dopants increased, the shape transition became more apparent (Appendix A). The Al:ZnO nanorods were formed by hydrothermal growth of Zn hydroxyl anions with substitutional doping of Al hydroxyl anions. The different valence states and sizes of the two anions induced unique needle shapes for Al:ZnO [23]. Both ZnO and Al:ZnO nanorods were spontaneously transformed to ZnTe and Al:ZnTe via Te^2−^ anion exchange reactions. The radii of ZnTe and Al:ZnTe became larger than those of ZnO and Al:ZnO, respectively, preserving the densities of nanorods in each film (Figure 2b,d). HR-TEM revealed the formation of ZnTe with a lattice spacing of 0.351 nm for zinc blend ZnTe(111) at the outer part of the nanorods (Figure 2e,f). 

### 3.2. Photoelectrocatalytic Syngas Production Using Al:ZnTe Nanorod Photocathode

Photoelectrochemical CO_2_ reductions were investigated in a conventional three-electrode configuration, using the developed ZnTe-based electrode as a working electrode, an Ag/AgCl reference electrode, and a graphite rod counter electrode, under simulated solar illumination (AM 1.5 G, 100 mW/cm^2^). To estimate the activity under dark and light conditions simultaneously, the photocurrent-potential (J-V) was measured under chopped illumination in a CO_2_-saturated KHCO_3_ electrolyte (Figure 3a). The photovoltage gain for CO_2_-to-CO reduction, the difference between the photocurrent onset potential of photocathodes, and the thermodynamic CO_2_/CO redox potential (E_onset_–E°_CO_2_/CO_) were clearly observed at more than 0.4 V for both ZnTe and Al:ZnTe. For the photocurrent density at the theoretical CO_2_/CO redox potential (E°_CO_2_/CO_), −0.11 V_RHE_ of the Al:ZnTe photocathode exhibited −1.1 mA/cm^2^, which was at least 3.5 times higher than that of the ZnTe photocathode. In the potential range of −0.4 to 0.3 V_RHE_, the Al:ZnTe photocathode exhibited enhanced photocurrent densities compared to the ZnTe photocathode. 

Let us consider the critical origin of the improvement of photocurrent densities due to Al doping on ZnTe. The first possibility is the enhancement of the number of photo-excited carriers resulting from the improvement of the light absorption properties of Al:ZnTe compared to ZnTe [24,25,26,27]. However, the light absorbance of Al:ZnTe was approximately similar (330–800 nm) to that of ZnTe. (Figure 3b) Considering the direct bandgap of Al:ZnTe (2.26 ± 0.02 eV) and ZnTe (2.24 ± 0.03 eV) determined by Kubelka-Munk relation using their absorbance, Al doping strategy had a negligible effect to increase the number of photo-generated carriers. 

Another possibility is the enhancement of photo-excited carrier separation due to Al doping, resulting from the decrease of resistance to charge transfer [28,29,30]. The Nyquist plots, prepared by measuring EIS, show that Al:ZnTe has a smaller semicircular radius than ZnTe [31] (Figure 3c). As the radius in the Nyquist plots corresponds to the impedance value, it indicates that Al:ZnTe has a relatively reduced impedance and improved charge transfer ability compared to pristine ZnTe. Therefore, the increased majority carriers and surface area by substitutional Al doping are considered to be the two main origins of photocurrent enhancement. 

To investigate the product distributions, photoelectrochemical CO_2_ reduction was performed on the photocathodes at −0.11 V_RHE_ which is thermodynamic CO_2_/CO redox potential. The ZnTe-based photocathodes produced H_2_ and CO via two-electron transfer using protons or dissolved CO_2_ in the electrolyte, but no additional byproduct was detected (Figure 4). The total product amounts, the sum of H_2_ and CO, of the Al:ZnTe photocathode were at least three times higher than those for ZnTe, which perfectly corresponded to increased photocurrents almost three times because of Al doping on ZnTe (Figure 3a). 

Furthermore, to investigate selectivity change resulting from Al doping, the ratio of H_2_/CO was compared using data collected from product distributions at −0.11 V_RHE_ (Figure 4c). ZnTe had a ratio of 5.2, whereas Al:ZnTe had a ratio of 5.0. The close ratio values indicate that the H_2_ production reaction is approximately five times more favorable than the CO_2_-to-CO reduction reaction on the surface of photoelectrodes for both ZnTe and Al:ZnTe as reported previously [16,17]. This suggests that both photoelectrodes provide an identical catalytic nature for electrochemical reactions and that Al dopants were ineffective in altering favorable catalytic reactions. 

To further study the possibility of switching major products during photoelectrochemical CO_2_ reduction, surface modification using Au electrocatalysts on Al:ZnTe (Al:ZnTe-Au) was conducted. The Au nanoparticles were physically deposited using an E-Beam evaporator and their atomic ratio was approximately 1% [13]. The J-V curve of Al:ZnTe-Au was identical to that of Al:ZnTe, indicating that the electron transfer rate remained unchanged after Au deposition (Figure 5a). 

In addition, to investigate product distribution differences due to Au coupling, the Al:ZnTe-Au photocathode underwent potential constant photoelectrolysis at −0.11 V_RHE_ and faradaic efficiencies for CO and H_2_ were determined based on the quantification results (Figure 5b,c). The imperfect total faradaic efficiencies observed for all ZnTe based photocathode are mainly due to the instability of ZnTe in an aqueous solution [16]. Al:ZnTe-Au exhibited dominant CO production, with FE_CO_ = 60% and minor H_2_ production with FE_H2_ = 15%. By contrast, Al:ZnTe showed limited CO production (FE_CO_ = 12%) with robust H_2_ evolution (60%). The Au nanoparticles can provide active sites for CO production and suppress H_2_ production owing to their intrinsically superior electronic nature for CO_2_ adsorption, CO_2_-to-CO conversion, and CO desorption using transferred protons and electrons, resulting in the major product switch [32,33,34]. The results demonstrate the availability of photoelectrochemical syngas production. Furthermore, the composition of the syngas—H_2_/CO—can be altered by modifying the light absorber using an electrocatalyst. 

## 4. Conclusions

Photoelectrochemical systems can provide an ideal method for direct syngas production via simultaneous CO_2_ and water reduction at ambient temperatures and pressures. This provides an alternative to the conventional process, requiring fossil fuels and natural gases at high temperatures and pressures. In this study, highly advanced Al-doped ZnTe nanorod photocathodes (Al:ZnTe) were developed for solar-energy-driven syngas production. This system has the following advantages: (1) a cost-effective solution process to prepare samples, (2) a nanorod array for facilitating photo-excited carrier separation, and (3) substitutional in situ Al doping on the site Zn while hydrothermally growing ZnO NW. Furthermore, a simple strategy was proposed for switching the major product to CO by suppressing competitive H_2_ evolution. Au electrocatalyst coupling can provide active sites for CO production. By the combination of all modifications, the advanced Al:ZnTe-Au photocathode represented −1.1 mA/cm^2^ at −0.11 V_RHE_ and the ratio of the H_2_:CO to 1:4, whereas pristine ZnTe exhibited −0.31 mA/cm^2^ with the syngas production ratio to 5:1. With further research advances, the developed system can be applied for solar-driven value-added chemicals production in practical.

## Figures and Tables

**Figure 1 materials-15-03102-f001:**
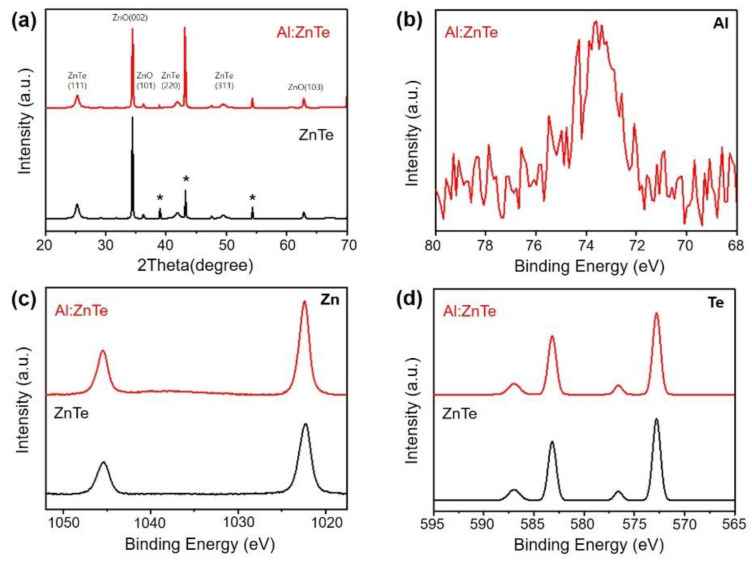
(**a**) XRD patterns, and XPS spectra for (**b**) Al 2p, (**c**) Zn 2p, and (**d**) Te 4f of Al:ZnTe and ZnTe. * indicates XRD patterns of FTO substrates.

**Figure 2 materials-15-03102-f002:**
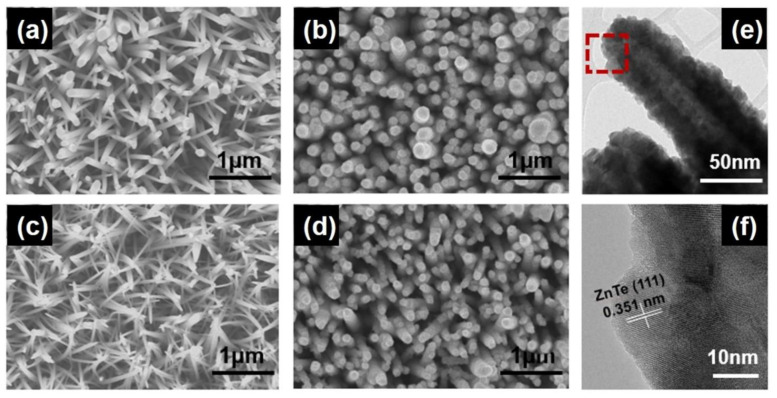
SEM images for (**a**) ZnO, (**b**) ZnTe, (**c**) Al:ZnO, and (**d**) Al:ZnTe. (**e**) HR-TEM results for Al:ZnTe, and (**f**) the magnified view of the region within the red box in (**e**).

**Figure 3 materials-15-03102-f003:**
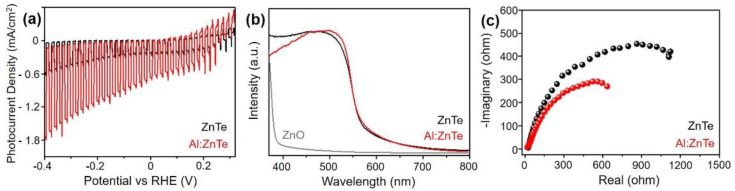
(**a**) J-V plot (scan rate of 5 mV/s), (**b**) absorbance, and (**c**) Nyquist plot of Al:ZnTe and ZnTe. All photoelectrochemical measurements were performed in 0.5 M CO_2_-saturated KHCO_3_ under simulated solar illumination (AM 1.5 G, 100 mW/cm^2^). The Nyquist plots were prepared by measuring electrochemical impedance spectroscopy at −0.11 V_RHE_.

**Figure 4 materials-15-03102-f004:**
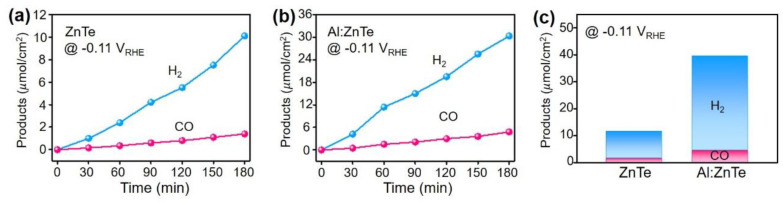
Time profiled products (H_2_ and CO) distribution using (**a**) ZnTe and (**b**) Al:ZnTe. (**c**) Comparison plot for accumulated products at −0.11 V_RHE_ measured for 3 h.

**Figure 5 materials-15-03102-f005:**
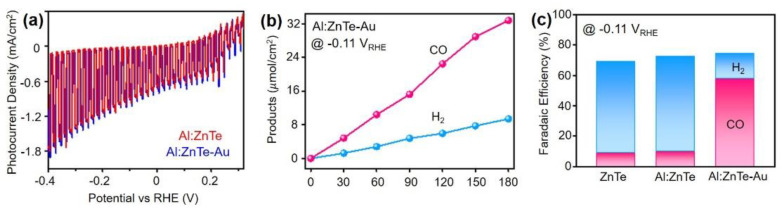
(**a**) J-V plot (scan rate of 5 mW/s) of Al:ZnTe and Al:ZnTe-Au. (**b**) Time-profiled product distribution. (**c**) Faradaic efficiency of each product of H_2_ and CO for ZnTe, Al:ZnTe, and Al:ZnTe/Au photocathodes at −0.11 V_RHE_.

## Data Availability

The data presented in this study are available upon request from the corresponding author.

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
