# Peer review of "Solar-Driven Syngas Production Using Al-Doped ZnTe Nanorod Photocathodes"

_materials, 2022, doi:10.3390/ma15093102_

Round 1

Reviewer 1 Report

It was my pleasure to review this work by Youn Jeong Jang and co-workers submitted to MDPI Materials. The authors have developed a novel functional material, Al-doped ZnTe, for photoelectrochemical syngas production. Moreover, the authors demonstrated that by adding a co-catalyst (Au), the ratio of reaction products in the resulting syngas mixture can be effectively varied. The presented work is clearly structured and presents a significant interest for the research community. However, I would recommend to address certain issues before this research can be published (major revision). The full report is attached.

Reviewer 2 Report

Journal: materials

Manuscript ID: materials-1683727

Title: Solar-driven syngas production using Al-doped ZnTe nanorod photocathodes

Comment on the manuscript:

In the current manuscript, the authors presented the potential use of Al-doped ZnTe nanorod as photocathodes materials for syngas production. Results demonstrated the applicability of the prepared photocatalyst through simultaneous photoelectrochemical CO2 and water reduction at ambient temperatures and pressures. However, in my opinion, this manuscript does not meet the standard of the journal in its current form due to several information went missing and can be considered for the publication in this journal only after the minor revision.  

Specific comments:

  1. The materials used/purity is missing in the experimental section.
  2. Authors should include the illustration on the experimental setup for clear understanding.
  3. Band gap structure “V vs NHE” is missing.
  4. The calculation for the band gap determination through Kubelka-Munk function is missing.
  5. Nitrogen adsorption-desorption isotherms and pore size distribution is missing.
  6. There should be a comparison between previous report/researches and author’s current work that employed ZnTe and its performance for the syngas production.
  7. Lifespan and stability of the photocatalysts. Authors need to include how the catalyst recovery is conducted prior to the next treatment cycle (for recyclability study). Will there be any leaching after the reaction completed?
  8. The conclusion must be supported by the data/results obtained.
  9. Some literatures reported recently in 2017-2022 should be cited and remove/replace the outdated references in the manuscript.

Reviewer 3 Report

A study titled Solar-driven syngas production using Al-doped ZnTe nanorod 2 photocathodes did well in terms of performance and efficiency.

Author Response

Thank you for your response to this work.

Round 2

Reviewer 1 Report

I would like to sincerely thank the authors for their revision and clarification they kindly provided. I am now convinced the manuscript can be published in its current form.